# Cefdet: Cognitive Effectiveness Network Based on Fuzzy Inference for Action Detection

## ABSTRACT

Action detection and understanding provide the foundation for the generation and interaction of multimedia content. However, existing methods mainly focus on constructing complex relational inference networks, overlooking the judgment of detection effectiveness. Moreover, these methods frequently generate detection results with cognitive abnormalities. To solve the above problems, this study proposes a cognitive effectiveness network based on fuzzy inference (Cefdet), which introduces the concept of "cognition-based detection" to simulate human cognition. First, a fuzzy-driven cognitive effectiveness evaluation module (FCM) is established to introduce fuzzy inference into action detection. FCM is combined with human action features to simulate the cognition-based detection process, which clearly locates the position of frames with cognitive abnormalities. Then, a fuzzy cognitive update strategy (FCS) is proposed based on the FCM, which utilizes fuzzy logic to re-detect the cognition-based detection results and effectively update the results with cognitive abnormalities. Experimental results demonstrate that Cefdet exhibits superior performance against several mainstream algorithms on public datasets, validating its effectiveness and superiority.

## CCS CONCEPTS

• **Computing methodologies** → **Activity recognition and understanding**.

## KEYWORDS

Multimedia content; Action detection; Fuzzy inference; Visual cognition; Feature fusion

## 1 INTRODUCTION

Multimedia is an essential branch of modern information technology, which comprehensively processes various media forms of information such as text, images, audio, and video through computers or other electronic devices. With the improvement of computer processing capabilities, the creation and sharing of multimedia content have been further promoted. Multimedia technology has driven the innovation of content presentation and provided a wealth of application scenarios for artificial intelligence research [1, 2].

Action detection is widely applied in multimedia analysis. It focuses on identifying specific human actions or behavior patterns

*MM '24, October 28-November 1, 2024, Melbourne, Australia*
© 2024 Copyright held by the owner/author(s). Publication rights licensed to ACM.
ACM ISBN 978-1-4503-XXXX-X/18/06
https://doi.org/XXXXXXX.XXXXXXX

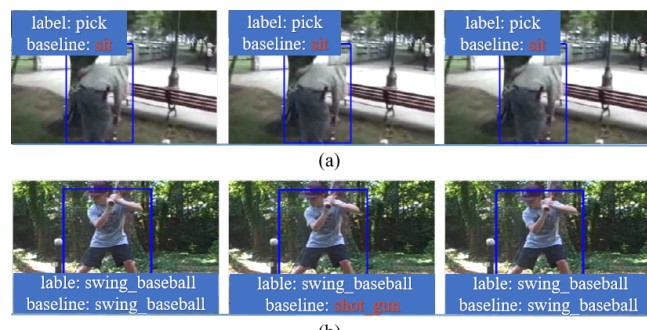

(a)

(b)

**Figure 1: Detection results with cognitive abnormalities in three consecutive frames. (a) is the false detection result of existing methods in highly similar actions, and (b) denotes the detection results of existing methods that do not conform to human action norms.**

from videos and can be combined with audio signals or other types of sensor data for multimedia analysis. In recent years, the accuracy of action detection has significantly improved with the advent of deep learning, particularly convolutional neural networks (CNNs) and recurrent neural networks (RNNs). Action detection is gradually playing a crucial role in various applications, including smart security, health monitoring, and human-computer interactions [3, 4].

However, existing methods face challenges in determining the effectiveness of detection results. Additionally, existing methods constantly generate detection results with cognitive abnormalities due to ineffective judgment. As shown in Figure 1 (a), actions such as "pick" and "sit" exhibit high similarity. Existing methods wrongly detect "pick" as "sit" while affecting the action detection of subsequent frames. In addition, as depicted in Figure 1 (b), action changes from "swing_baseball" to "shot_gun" do not conform to human action norms. Existing methods detect two unrelated actions as adjacent and recognize one or several consecutive frames as a complete action, which is inconsistent with human cognition. These problems hinder the further application of action detection methods.

Therefore, this study proposes a cognitive effectiveness network based on fuzzy inference (Cefdet). It introduces the concept of "cognition-based detection" and combines features in human action with fuzzy inference to simulate the cognition-based detection process. The effectiveness of the detection results is accurately determined using cognition-based detection. Moreover, the detection results are divided based on their effectiveness, and fuzzy logic is employed to dynamically update the detection results with low-level cognition, which repairs the detection results with cognitive abnormalities.

The contributions of this study are summarized as follows:

- This paper proposes the FCM for detecting the effectiveness of frames. It takes the features of human actions, including the confidence of each frame, the correlation between adjacent frames, and the position score of each frame, as inputs to the fuzzy inference system. This simulates a cognition-based detection process to obtain effectiveness, thus accurately locating the position of action frames with cognitive abnormalities.
- In this study, the FCS based on the above FCM is used to update the results dynamically. It divides cognition-based detection results into high and low levels. Subsequently, the local features of the frames with low-level cognition are weighted and fused through frames with high-level cognition for re-detection. The frames before and after re-detection are dynamically updated based on fuzzy logic to obtain optimal results, effectively repairing the detection results with cognitive abnormalities.
- Experimental results show that Cefdet performs better on public datasets than existing methods. Furthermore, Cefdet promotes the application of fuzzy inferences in computer vision perception.

## 2 RELATED WORK

### 2.1 Video understanding

Video understanding includes identifying the activities that occur during the editing process. Typically, the time span for editing is a few seconds, and there is only one annotation. Most early video comprehension methods [5, 6] utilized 2D image CNNs and introduced long-short-term memory (LSTM) [7] to learn video time structures. Subsequently, various 3D CNNs [8] are proposed for video understanding because they process the entire video clip as input rather than treat it as a frame sequence [9]. Due to the scarcity of tagged video datasets, several researchers have relied on a model pre-trained on ImageNet and used it as the backbone for video feature extraction. Some studies [10, 11] decoupled 3D convolution into 2D space and 1D time kernels to reduce the model size. Two-stream networks [12] constitute another widely employed video understanding method. Due to its ability to handle only a small portion of the input frames, it achieved a superior balance between accuracy and complexity. Cheng et al. [13] proposed a new method to recover the intermediate features between two sparse samples and adjacent video frames, achieving significant results.

### 2.2 Action detection

Action detection has received increasing attention from researchers as an essential technology for video understanding [14]. Owing to the potential of multiple actions within each frame, it is necessary to detect the actions of individual entities within the current frame rather than categorizing the entire video into a single class. Inspired by deep CNNs for object detection [15], methods based on action detection typically apply 2D position anchors or offline object detectors to keyframes to locate human subjects. Subsequently, they focused on improving action detection and incorporating temporal patterns by leveraging the optical flow for additional flow fusion. Some methods [16, 17] have applied 3D convolutional networks to capture temporal information for identifying actions and have

achieved excellent results. Feichtenhofer et al. [14] proposed a slow network that can better capture spatiotemporal information.

Recent research on spatiotemporal action detection has emphasized modeling the interaction between classified individuals and their environment. Recent methods [18, 19] proposed explicitly modeling the relationships between actors and objects. A dual-mode [20] interaction structure is constructed based on human posture, hands, and objects, effectively improving the accuracy of action detection.

### 2.3 Fuzzy inference

Fuzzy inference is widely used in computer vision applications. It is a core component of fuzzy logic and involves the application of fuzzy sets and rules to input data to obtain fuzzy output results. Fuzzy inference performs better in handling uncertain information and can more reasonably describe the actual needs and achieve intelligent control. Several studies based on fuzzy inference have been proposed. For example, an improved fuzzy clustering-based classifier [21] employs $L_2$ regularization to mitigate overfitting. This classifier demonstrated remarkable performance in various classification tasks. Eyoh et al. [22] introduced an interval-based intuitionistic fuzzy system with membership and non-membership functions for identification and prediction. Multi-layer interval Type-2 fuzzy limit machine learning (ML-IT2-FELM) [23] identify walking activities and used wearable sensors to determine gait. Fuzzy learning is utilized to obtain appropriate results for the activities. Cao et al. [24] integrated interval Type-2 fuzzy sets into a fuzzy rough neural network to forecast stock time series. In addition, other studies [25, 26] introduced a common quadratic Lyapunov function to analyze the stability and design controllers for fuzzy closed-loop systems.

## 3 PROPOSED METHOD

### 3.1 Overall framework

Due to the uncertainty of daily human activities, existing algorithms cannot effectively determine the effectiveness of their detection results. Moreover, the lack of effective judgment frequently leads to detection results with cognitive abnormalities in existing algorithms. To solve these problems, this study introduces the concept of "cognition-based detection" into action detection. First, fuzzy inference is employed to simulate the cognition-based detection process by combining human action features to accurately locate the positions of frames with cognitive abnormalities. Then, the local features of the frames with cognitive abnormalities are dynamically fused based on fuzzy logic for re-detection. The effectiveness before and after re-detection is used to repair detection results inconsistent with cognition.

The overall framework of Cefdet is illustrated in Figure 2. Initially, the FCM is designed after the video frames pass through a network. It evaluates the effectiveness of each frame by considering the confidence, correlation between adjacent frames, and position score, dividing the frames into low and high levels. Subsequently, the FCS is proposed to update the frames with low-level cognition. Specifically, it constructs correlated sequences using frames with high-level cognition. The features are then weighted based on the

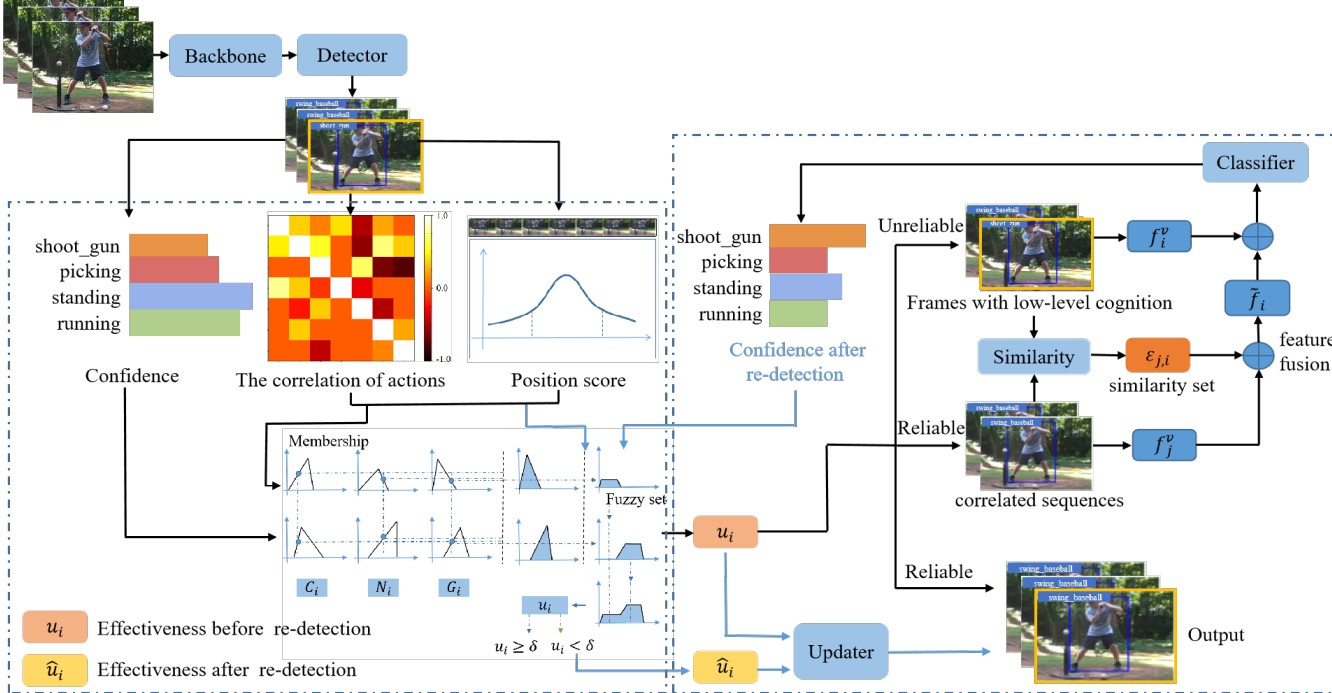

**Figure 2: The overall framework of Cefdet for action detection. The left module is the Fuzzy-driven cognitive effectiveness evaluation module, abbreviated as FCM, and on the right is the Fuzzy cognitive update strategy, termed FCS.**

similarity between the correlated sequences and frames with low-level cognition for fusion, which is utilized for re-detection. Finally, the decision to update the results is made based on the effectiveness of the action frames before and after the re-detection.

## 3.2 Fuzzy-driven cognitive effectiveness evaluation module (FCM)

Due to the lack of effective judgment of detection results, existing action detection algorithms frequently obtain detection results with cognitive abnormalities in complex scenes. The use of confidence scores to evaluate the effectiveness of an action frame is not comprehensive, and more factors should be considered. Therefore, this study proposes the FCM that cooperatively judges each frame's effectiveness by combining each frame's confidence, the correlation between adjacent frames, and the position score of each frame with a fuzzy inference system. Fuzzy logic is employed to simulate the cognition-based detection process to accurately determine the effectiveness of each frame and locate the positions of frames with cognitive abnormalities. FCM is a fuzzy inference engine with four components: feature quantization, fuzzification, fuzzy inference, and defuzzification. Brief descriptions of each component are provided below.

**Feature quantification.** The objective of fuzzy inference is to assess the effectiveness of the frames after detection. Therefore, it is essential to construct feature vectors of the frames as inputs for fuzzy inference. For the input video sequence $I_i$, the feature vector consists of diverse feature values, including the confidence of each frame $C_i$, correlation between adjacent frames $N_i$, and position score of each frame $G_i$.

1. Confidence. As an important evaluation criterion, the confidence of the action detection network is expressed in the following formula.

$$C_i = \varphi(I_i), \tag{1}$$

where $\varphi(*)$ refers to the action detection network in [20], which is tasked to predict the true value from $I_i$.

2. Correlation between adjacent frames. In video sequences, action sequences with short intervals have rich correlations, and the evolution from one action to another is a gradual process. It is possible to determine whether the subsequent action is reasonable based on a previous action. Therefore, the positions of action frames that do not conform to human action norms are accurately located using this correlation. This study employs normalized pointwise mutual information (NPMI) to measure the correlation between actions. NPMI is defined as follows:

$$N_i = \left(ln\frac{P(K_{I_{i-1}}, K_{I_i})}{P(K_{I_{i-1}}) p(K_{I_i})}\right) / \left(-\ln P(K_{I_{i-1}}, K_{I_i})\right), \tag{2}$$

where $K_{I_i}$ refers to the action category of the i-th frame; $P(K_{I_i})$ is the probability of $K_{I_i}$; and $P(K_{I_{i-1}}, K_{I_i})$ denotes the probability of a joint distribution between actions. The value of $N_i$ is in the range [-1, 1], where $N_i = 1$ represents a high degree of correlation

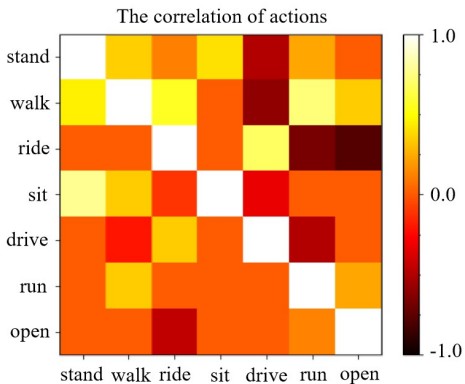

**Figure 3: The correlation between different actions. The correlations between all actions are normalized to be between [-1, 1], with darker colors indicating lower correlations.**

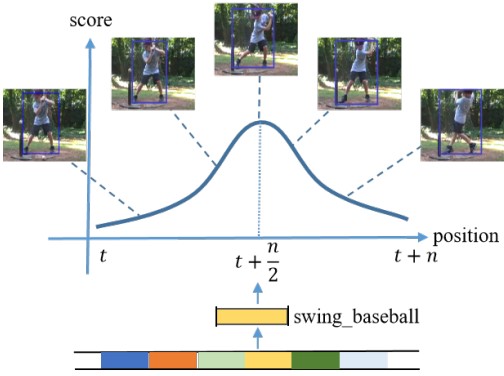

**Figure 4: Position score of frames in action sequences. In a complete action sequence, the closer to the center the frame is, the more reliable it is.**

**Table 1: Examples of partial fuzzy rules.**

| Rule | $C$ | $N$ | $G$ | $U$ |
|------|-----|-----|-----|-----|
| $R_1$ | $NB$ | $PB$ | $NB$ | $PB$ |
| $R_2$ | $NB$ | $ZO$ | $NS$ | $PS$ |
| $R_3$ | $NS$ | $NS$ | $ZO$ | $ZO$ |
| $R_4$ | $PS$ | $NB$ | $PB$ | $NS$ |
| $R_5$ | $NS$ | $ZO$ | $ZO$ | $PS$ |
| $R_6$ | $NS$ | $NS$ | $NB$ | $ZO$ |
| $R_7$ | $PB$ | $NB$ | $PB$ | $NS$ |
| $R_8$ | $NS$ | $NS$ | $ZO$ | $NB$ |

between the two actions, $N_i = 0$ indicates independence between the two actions, and $N_i = -1$ denotes that the two actions have never appeared simultaneously. Figure 3 shows the correlation between the different actions calculated by the NPMI.

3. Position score. Action detection is the process of observation and refinement with a clear time boundary between the beginning and end of an action. The position of the current action frame is noted. If the frame is at the center of the current action, it is considered highly reliable. By contrast, if the frame is at the boundary position of the current action, the reliability of the frame requires further detection. In this study, the reliability of a frame is determined by comparing its corresponding position within a complete action. For action category $K_{I_t}$ at frame $t$, if $K_{I_{t-1}} \neq K_{I_t}, K_{I_t} = K_{I_{t+1}} = \cdots = K_{I_{t+n}}$ and $K_{I_t} \neq K_{I_{t+n+1}}$, then $[I_t, \cdots, I_{t+n}]$ can be considered as a complete action. The reliability of this frame is defined as follows:

$$G_i = \frac{1}{\sigma\sqrt{2\pi}}e^{-\frac{1}{2}z^2}, \quad (3)$$

$$z = \frac{i-\mu}{\sigma}, \ i \in [t, t+n], \quad (4)$$

where $G_i$ is the position score of the i-th frame; $\mu$ and $\sigma$ are the mean and standard deviation of the Gaussian distribution, respectively; $\mu = t + \frac{n}{2}$, and $\sigma = \sqrt{\frac{\sum_{i=t}^{t+n}(i-\mu)^2}{n}}$. Figure 4 shows the position scores of the frames in the complete action sequence.

**Fuzzification.** The five-level fuzzification rule $k = \{NB, NS, ZO, PS, PB\}$ is employed to determine the degree of membership of the feature vector to a fuzzy set: negative big ($NB$), negative small ($NS$), zero ($ZO$), positive small ($PS$), or positive big ($PB$). $h_i = \{C_i, N_i, G_i, U_i\}$ is the training set of fuzzy inference, where $U_i = \mu_1 C_i + \mu_2 N_i + (1 - \mu_1 - \mu_2) G_i$, the universe is defined as $P = \{P(h_i)\}$, $\mu_{k_P}(h_i)$ indicates the membership degree of $h_i$ to the set $k_P$, and $F_{h_i} = \{\mu_{k_P}(h_i)\}$ reports that $h_i$ belongs to the fuzzy set of universe $P$, as illustrated below:

$$F_{h_i} = \left\{\frac{\mu_{k_{P(h_i)}}(h_i)}{h_i}\right\}. \quad (5)$$

Thus, the fuzzy control rule is obtained by extracting the front parts $F_{C_i}$, $F_{N_i}$, and $F_{G_i}$ and the rear part $F_{U_i}$.

**Fuzzy inference.** Because the fuzzy inference system had three input variables, driving 125 fuzzy rules. Table 1 shows examples of fuzzy rules. The fuzzy set generated by each fuzzy rule $\omega_{i,j}$ is obtained by performing the combination of membership degrees on the front parts of each rule, where $j \in [1, 2, \cdots, w]$ is the number of active rules. Subsequently, the conclusion of fuzzy inference $\hat{U}_i$ is obtained through a disjunction operation. The formula is as follows:

$$\omega_{i,j} = if\left(C_i \text{ is } F_{C_i}\right) \text{ and } \left(N_i \text{ is } F_{N_i}\right) \text{ and } \left(G_i \text{ is } F_{G_i}\right) \text{ then } \left(U_i^j \text{ is } F_{U_i}^j\right), \quad (6)$$

$$\widehat{U}_i = \bigvee_{j=1}^{w} \omega_{i,j}, \quad (7)$$

where $U_i^j$ is the j-th element in the output universe $P(U_i)$, and $F_{U_i}^j \in k_{P(U_i)}$ denotes any fuzzy set on $P(U_i)$.

**Defuzzification.** During defuzzification, the centroid method is selected to obtain the output. The abscissa value corresponding to the membership function curve of the fuzzy set and the center of the area surrounded by it is the effectiveness of frame $u_i$, which can be defined as follows:

$$u_i = \frac{\sum_{j=1}^{w} U_i^j \mu_{k_{P(U_i)}} \left( U_i^j \right)}{\sum_{j=1}^{w} \mu_{k_{P(U_i)}} \left( U_i^j \right)}, \tag{8}$$

where $\mu_{k_{P(U_i)}} \left( U_i^j \right)$ is the membership degree of $U_i^j$ to set $k_{P(U_i)}$, and $P\left(U_i\right) = \left\{ U_i^1, U_i^2, \cdots, U_i^j \right\}$.

Figure 5 shows the overall framework of FCM. First, the corresponding feature vectors are extracted from the action detection and are fuzzified into membership degrees of different sets in the respective universe. Fuzzy logic is then employed to aggregate the active fuzzy rules. Finally, the effectiveness of the frame is obtained through defuzzification.

The FCM evaluates the effectiveness of each frame by incorporating multiple perspectives. This aids in the accurate localization of the detection results with cognitive abnormalities and facilitates the subsequent stages of further detection.

### 3.3 Fuzzy cognitive update strategy (FCS)

The detection performance for each frame in a video sequence depends on the effectiveness of the surrounding action frames. The poor effectiveness of the surrounding action frames often leads to noise interference in detecting the target frame. Therefore, frames are divided into low and high levels based on their effectiveness. Frames with high-level cognition are considered reliable predictions, whereas the remaining frames require further detection as follows:

$$D_A \leftarrow \bigcup \{I_i | u_i \geq \delta\}, \ D_N \leftarrow I - D_A . \tag{9}$$

Subsequently, a storage space is constructed to store correlated sequences composed of frames with high-level cognition, and the features of the correlated sequences are combined to re-detect frames with low-level cognition. Subsequently, the results before and after re-detection are dynamically updated based on fuzzy logic, improving action detection performance. Specifically, this study proposes a fuzzy cognitive update strategy that includes three main parts: constructing correlated sequences, re-detecting frames with low-level cognition, and updating frames with low-level cognition.

**The construction of correlated sequences.** Frames with high-level cognition are used to construct correlated sequences that are applied in subsequent re-detection processes. As action detection continues, the size of the correlated sequences gradually increases. The matching of the frame with low-level cognition and all the frames with high-level cognition significantly affected the detection speed. Therefore, for a specific frame with low-level cognition $I_i$, only video frames within a certain neighboring range are employed to construct the correlated sequence $T_i$.

$$T_i \leftarrow \bigcup \left\{ I_j \middle| I_j \in D_A, j \in (i - \lambda, i + \lambda) \right\}, \tag{10}$$

where $\lambda$ is a hyperparameter used to determine the number of correlated sequences around frames with low-level cognition.

**Re-detection of frames with low-level cognition.** The correlated sequences provide rich contextual clues to determine the category of the current frame. It also contains noise that affects the determination of subsequent frame categories. Therefore, the core idea of re-detection is to simulate local evolution by dynamically

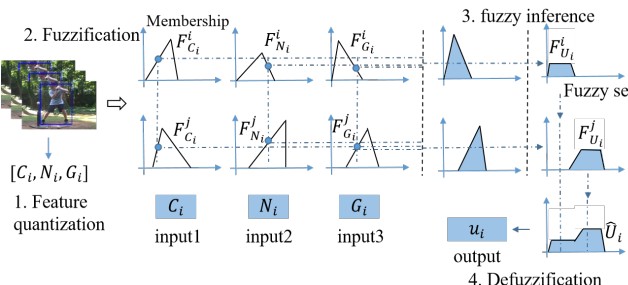

**Figure 5: Illustration of the FCM, which consists of four components: feature quantification, fuzzification, fuzzy inference and defuzzification.**

aggregating the features of frames with low-level cognition and their related sequences.

Specifically, we extract features based on the backbone network in [27]. For the extracted feature $f_j$ of the j-th frame image in the correlated sequence $T_i$, $f_j$ is converted into the key and value spaces, where the former is responsible for comparing similarities, and the latter can be utilized for feature aggregation. It is expressed as:

$$f_j^k = \Phi^k \left( f_j \right), \ f_j^v = \Phi^v \left( f_j \right), \tag{11}$$

where $\Phi^k$ and $\Phi^v$ denote the key mapping convolutional layer and value mapping convolutional layer in the re-detection network, respectively.

Then, the cosine similarity is calculated to measure the similarity between the feature of the correlated sequence $f_j^k$ and the feature of the frame with low-level cognition $f_i^k$, as follows:

$$\varepsilon_{j,i} = \frac{f_j^k \cdot f_i^k}{\left\| f_j^k \right\| \cdot \left\| f_i^k \right\|}, \tag{12}$$

where $\varepsilon_{j,i}$ refers to the similarity between the j-th frame of the correlated sequence $f_j^k$ and $f_i^k$. The similarity set $\left[ \varepsilon_{1,i}, \cdots, \varepsilon_{j,i} \right]$ is obtained from above, $j \in (1, 2, \cdots, z)$ is the number of frames in the correlated sequence, and softmax is employed to normalize and obtain the attention mask $\{\hat{\varepsilon}\}$. The feature values in the correlated sequence are aggregated to obtain features with high-level cognition $\tilde{f}_i$:

$$\tilde{f}_i = \sum_{j=1}^{z} \hat{\varepsilon}_{j,i} \cdot f_j^v. \tag{13}$$

Then, the features of the frames with low-level cognition $f_i^v$ and high-level cognition $\tilde{f}_i$ are used to obtain the predicted classification score $H_i$ after re-detection.

$$H_i = \Omega(\tilde{f}_i, f_i^v), \tag{14}$$

where $\Omega(\cdot, \cdot)$ is the classifier of the re-detection network, $H_i = \left( C^{K_1}, \cdots, C^{K_h} \right)$ indicates the set of classification scores after re-detection, and $C^{K_h}$ represents the predicted score of $K_h$. The infinite norm of the classification is considered the confidence $\hat{C}_i$ of the re-detection.

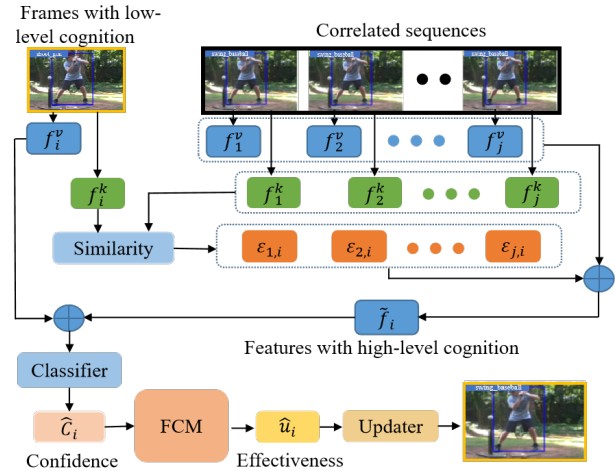

**Figure 6: Illustration of the FCS. FCS measures the similarity between features in correlated sequences and frames with low-level cognitive abilities, and fuses the features to obtain features with high-level cognitive abilities.**

$$\hat{C}_i = \|H_i\|_\infty. \tag{15}$$

**Update frames with low-level cognition.** From the above steps, confidence $\hat{C}_i$ after re-detection is obtained. However, updating the results is unreliable when only confidence is used after re-detection. Therefore, it is necessary to replace $C_i$ with $\hat{C}_i$ as the input for fuzzy inference and use Equations (5)-(8) to obtain the effectiveness $\hat{u}_i$ after the re-detection. The result is dynamically updated based on the effectiveness before and after re-detection.

$$u_{opt}^i = \max\left(\hat{u}_i, u_i + \tau\right), \tag{16}$$

where $u_i$ is the effectiveness of the initial frame and $\hat{u}_i$ refers to the effectiveness of the frame after re-detection. $\tau$ denotes a predefined threshold. Because the initial frame is the foundation of the detection process, the result's update occurs only when the combined effectiveness of the initial frame and the predefined threshold is lower than the effectiveness obtained after re-detection.

Figure 6 shows the overall process of the FCS. First, the correlated sequences of frames with high-level cognition are constructed for further detection. The similarity between the features in the correlated sequence and frames with low-level cognition is then measured, and the features are fused to obtain features with high-level cognition. The features of frames with low-level cognition and correlated sequences are jointly considered for re-detection. Fuzzy logic is then employed to dynamically update the detection results with low-level cognition before and after re-detection.

## 4 EXPERIMENTAL RESULTS AND ANALYSIS

To evaluate the superiority of Cefdet for action detection, experiments are conducted using JHMDB [28] and UCF101-24 [29]. The following introduces the JHMDB and UCF101-24 datasets and describes their implementation details. The experimental results of Cefdet are then quantitatively and qualitatively discussed to demonstrate its effectiveness.

### 4.1 Datasets

**JHMDB** [28] is a benchmark dataset for action detection. It consists of a video composed of 928 temporary clips from 21 different action categories. The dataset involves fine-grained actions and subtle temporal cues, requiring precise temporal localization for accurate detection.

**UCF101-24** [29] is derived from the UCF101 dataset and focuses on 24 action categories consisting of 3207 videos representing various human activities such as walking, jogging, basketball, and dancing. The videos exhibited variations in viewpoint, scale, background clutter, and lighting conditions, posing challenges for accurate action detection.

### 4.2 Implementation details

**Object detector:** The experiment extracted keyframes from each video in the JHMDB and UCF101-24 datasets and used the character bounding boxes detected in [30] for inference. The target detector adopted a Faster RCNN [31] in the ResNet-50-FPN backbone. The model is pretrained on ImageNet [32] and fine-tuned on MSCOCO [33].

**Backbone:** SlowFast [14] is the video backbone. The experiment is instantiated using SlowFast and ResNet-50 pretrained on Kinetics700 [34].

**Training and evaluation:** The model is trained for 7k iterations on the JHMDB dataset, where the first 700 iterations are for linear preheating. SGD is used as the optimizer with a batch size of 8. Similarly, there are 50k iterations of training on the UCF101-24 dataset, with linear preheating applied in the first 1k iterations. The learning rate is 0.0002, which is 10 times less in iterations of 25k and 35k. The hyperparameters $\mu_1$, $\mu_2$, and $\mu_3$ are 0.6, 0.2, and 0.2, respectively.

### 4.3 Quantitative analysis

To verify the effectiveness of Cefdet in action detection, we compared it with state-of-the-art (SOTA) methods using the JHMDB and UCF101-24 datasets. The frame mean Average Precision (mAP) with an intersection over union (IoU) threshold of 0.5 is used as the evaluation metric, and the experimental results are shown in Table 2 and 3.

The mAP of SOTA reached 82.9% and 84.8% for the JHMDB and UCF101-24 datasets, respectively. Although the SOTA algorithm exhibited advanced performance on the JHMDB and UCF101-24 datasets, its detection accuracy for high-similarity actions remained poor. Moreover, it is difficult for a detector to explore the relationship between continuous actions, resulting in detection results that do not conform to human action norms. Repairing these detection results with cognitive abnormalities is a challenge in action detection because of the lack of effective judgment of the detection results.

However, Cefdet achieved mAP values of 84.0% and 85.0% on the JHMDB and UCF101-24 datasets, respectively, which provided gains of 1.1% and 0.2%, respectively, compared with SOTA. Cefdet utilizes fuzzy inference to simulate a cognition-based detection

**Table 2: Experimental results using the JHMDB dataset based on mAP. Cefdet exhibits excellent performance on both frame mAP and video mAP.**

| Model | Input | f@0.5 | v@0.2 | v@0.5 |
|---|---|---|---|---|
| MOC(2020)[35] | V+F | 70.8 | 77.3 | 77.2 |
| AVA(2018)[16] | V+F | 73.3 | - | 78.6 |
| PCSC(2019)[36] | V+F | 74.8 | 82.6 | 82.2 |
| HISAN(2019)[37] | V+F | 76.7 | 85.9 | 84.0 |
| ACRN(2018)[17] | V+F | 77.9 | - | 80.0 |
| CA-RCNN(2020)[38] | V | 79.2 | - | - |
| WOO(2021)[39] | V | 80.5 | - | - |
| TubeR(2022)[40] | V+F | - | 87.4 | 82.3 |
| SE-STAD(2023)[41] | V | 82.5 | - | - |
| MCA-SVMM(2023)[42] | V | 74.9 | 82.1 | 81.8 |
| HIT(2023)[20] | V | 82.9 | 88.5 | 87.2 |
| **Ours** | V | **84.0** | **90.4** | **89.3** |

**Table 3: Experimental results on the UCF101-24 dataset based on mAP. Cefdet surpasses SOTA methods.**

| Model | Input | f@0.5 | v@0.2 | v@0.5 |
|---|---|---|---|---|
| HISAN(2019)[37] | V+F | 73.7 | 80.4 | 49.5 |
| MOC(2020)[35] | V+F | 78.0 | 82.8 | 53.8 |
| AVA(2018)[16] | V | 76.3 | - | - |
| AIA(2020)[18] | V | 78.8 | - | - |
| PCSC(2019)[36] | V+F | 79.2 | 84.3 | 61.0 |
| TubeR(2022)[40] | V+F | 83.2 | 83.3 | 58.4 |
| ACAR(2021)[19] | V | 84.3 | - | - |
| CycleACR(2023)[43] | V | 84.7 | - | - |
| MCA-SVMM(2023)[42] | V | 79.3 | 83.4 | 54.6 |
| HIT(2023)[20] | V | 84.8 | 88.8 | 74.3 |
| **Ours** | V | **85.1** | **89.3** | **75.1** |

**Table 4: Experimental results of representative actions in the JHMDB dataset (mAP).**

| Action | HIT(%) | Ours(%) | Gap(%) |
|---|---|---|---|
| Shoot_ball | 71.2 | 83.4 | +12.2 |
| Sit | 52.3 | 56.5 | +4.2 |
| Stand | 26.1 | 33.3 | +7.2 |
| Walk | 78.6 | 89.1 | +10.5 |
| Climb_stairs | 89.2 | 91.3 | +2.1 |
| Throw | 75.5 | 78.1 | +2.6 |

process, effectively determining the effectiveness of frames and accurately locating frames with cognitive abnormalities. Furthermore, each frame is divided into frames with high and low-level cognition based on their effectiveness. The features of the frames with high-level cognition are refined to assist in the re-detection of frames with low-level cognition. Fuzzy logic is then employed

**Table 5: The ablation experiment in the JHMDB dataset.**

| FCM | FCS | mAP |
|---|---|---|
| | | 82.9 |
| ✓ | | 83.8 |
| | ✓ | 83.2 |
| ✓ | ✓ | **84.0** |

**Table 6: Results of fuzzy cognitive update under different thresholds using confidence and effectiveness as evaluation criteria in the JHMDB dataset (mAP).**

| Threshold | 0.1 | 0.2 | 0.3 | 0.35 | 0.4 | 0.5 |
|---|---|---|---|---|---|---|
| Confidence | 83.03 | 83.13 | **83.22** | 83.20 | 83.16 | 83.11 |
| Effectiveness | 83.37 | 83.56 | 83.79 | **83.97** | 83.85 | 83.76 |

to dynamically update the detection results with low-level cognition before and after re-detection. This reduced the impact of noise and misjudgment, effectively repaired the detection results with cognitive abnormalities, and improved the robustness of the model.

The results in Table 4 indicate that the detection accuracy of action categories such as "shoot_ball", "walk", and "stand" is poor on the JHMDB dataset. This indicates that it is difficult to distinguish between similar actions using existing methods. However, Cefdet exhibited excellent performance in these action categories, with improvements of 12.2%, 10.5%, and 7.2%, respectively, compared with the SOTA algorithm. This improvement is attributed to evaluating the effectiveness of the fuzzy inference system. It simulates a cognition-based detection process and accurately identifies the location of the detection results with cognitive abnormalities. Subsequently, the detection results are divided into frames with high and low-level cognition based on their effectiveness. Frames with low-level cognition are re-detected by similarity-weighted fusion of local features, and frames with higher-level cognition select the results before and after re-detection. This effectively repairs detection results with cognitive abnormalities.

The above experimental results demonstrate that Cefdet achieves SOTA performance on the JHMDB and UCF101-24 datasets. This fully reflects Cefdet's superiority and robustness.

## 4.4 Ablation studies

In this section, ablation experiments are conducted on JHMDB to investigate the effectiveness of FCM and FCS. The frame mAP with an IoU threshold of 0.5 is used as an evaluation metric, and the experimental results are shown in Table 5 and 6.

Notably, when FCM is removed, the confidence score for action detection is selected as the evaluation criterion. The obtained mAP is 83.2%, which is a reduction of 0.8%, indicating that effectiveness had a more significant effect than confidence. Table 6 presents the results of using confidence and effectiveness as the evaluation criteria. The effectiveness results as the evaluation criterion are consistently higher than those of using confidence under different thresholds, and the highest mAP of 83.97 is achieved at a threshold of 0.35, demonstrating the superiority of FCM and indicating

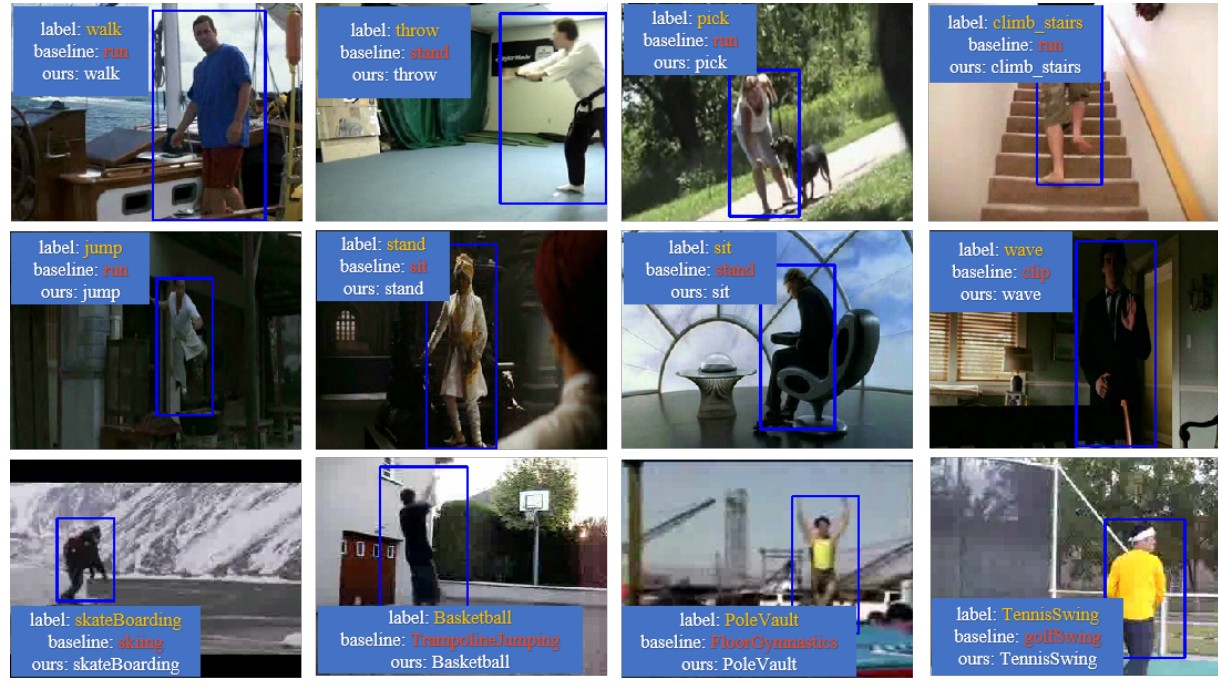

**Figure 7: Visualization results of Cefdet on the JHMDB and UCF101-24 datasets. In the images, yellow indicates the true annotation, while red and white represent the detection results of holistic interaction transformer (HIT) and Cefdet, respectively. All detection boxes are shown in blue.**

that cognition-based detection achieves accurate localization of the positions of the detection results with cognitive abnormalities.

The experimental results are updated based on confidence before and after re-detection after removing the FCS. The result is a 0.2% decrease, demonstrating that using only confidence scores to update the experimental results is unreasonable. Cognition-based detection exhibits excellent performance in handling nonlinear, time-varying, and unclear information, effectively improving the accuracy of detecting cognitive abnormalities.

The results demonstrated that both the FCM and FCS effectively enhanced the accuracy of action detection in complex scenarios.

### 4.5 Visualization

Figure 7 illustrates the visual results of Cefdet on the JHMDB and UCF101-24 datasets, which reflects the superiority of Cefdet in repairing the detection results with cognitive abnormalities. This improvement is attributed to the combination of human action features and fuzzy inference to simulate a cognition-based detection process. It accurately localizes the detection results of cognitive abnormalities. Furthermore, fuzzy logic is utilized to aggregate the local features of frames with low-level cognition for re-detection, and the detection results are dynamically updated based on the effectiveness of the frames before and after re-detection. This effectively repairs the detection results with cognitive abnormalities.

These results indicate that Cefdet performs superiorly in these challenging situations and is more suitable for action detection.

## 5 CONCLUSION AND FUTURE WORK

This study proposed Cefdet, which introduced the concept of "cognition-based detection" into action detection to simulate human cognition. First, the FCM is designed to evaluate the effectiveness of the frames. It extracts various action features from video sequences and inputs them into fuzzy inference. The cognition-based detection process is simulated using a fuzzy system to obtain effectiveness, which accurately locates the position of action frames with cognitive abnormalities. Then, the FCS is proposed based on the FCM to update the results dynamically. It constructs related sequences by frames with high-level cognition and performs a weighted fusion of features by fuzzy inference for re-detection. Subsequently, the results before and after re-detection are dynamically updated, which effectively repairs the detection results with cognitive abnormalities. Experiments on public datasets proved that Cefdet achieves superior performance and promotes the further application of fuzzy inference in action detection.

In future work, our objective will be to incorporate interval type-2 fuzzy set theory into developing a robust and real-time action detection framework and further facilitate the application and advancement of fuzzy theory in action detection.

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
