# OpenReview forum: "Cefdet: Cognitive Effectiveness Network Based on Fuzzy Inference for Action Detection"
_acmmm.org/ACMMM/2024/Conference — MM2024 Poster_

### Official Review · Reviewer_yHqX · 2024-05-13

**Rating:** 4
**Confidence:** 3

**Summary:**

The manuscript proposes a cognitive validity network (Cefdet) based on fuzzy reasoning and introduces the concept of "cognition-based detection" to simulate human cognition. First, a fuzzy-driven cognitive validity assessment module (FCM) is proposed to introduce fuzzy reasoning into action detection. Then, a fuzzy cognitive update strategy (FCS) is proposed based on FCM, which uses fuzzy logic to re-detect cognitive-based detection results and effectively update cognitive abnormal results.

**Strengths:**

1.The manuscript introduces the cognitive detection process and fuzzy inference process into the action detection task, which is very interesting and innovative.
2.The method proposed in the manuscript achieves advanced results on two data sets.

**Limitations:**

1.The JHMDB or UCF101-24 used in the manuscript experiments are two small datasets in the action detection task, and it is curious to see how the effectiveness of the method proposed in the manuscript performs on datasets of large size or many categories.
2.I'm curious about the connection between fuzzy rules and actions. The fuzzy inference process seems to have many rules. How to reason about actions based on the rules?
3.The manuscript does not mention the loss function optimization process of the method.
4.How is the Updater in Figure 2 implemented? This is not explicitly done in the method.
5.The manuscript focuses action detection, and the methods are updated by detecting abnormal actions. Some work related to these is missing, such as:
[1]Hierarchical scene normality-binding modeling for anomaly detection in surveillance videos[C]//Proceedings of the 30th ACM international conference on multimedia. 2022: 6103-6112.

[2] Egocentric Early Action Prediction via Multimodal Transformer-Based Dual Action Prediction. IEEE Trans. Circuits Syst. Video Technol. 33(9): 4472-4483 (2023)

[3] A Knowledge-Based Hierarchical Causal Inference Network for Video Action Recognition[J]. IEEE Transactions on Multimedia, 2024.

[4] Eval: Explainable video anomaly localization. Proceedings of the IEEE/CVF Conference on Computer Vision and Pattern Recognition. 2023.

**Suitability:**

3

---

### Official Review · Reviewer_bpGh · 2024-05-25

**Rating:** 4
**Confidence:** 3

**Summary:**

The paper proposes a cognitive effectiveness network for action detection based on fuzzy inference (Cefdet), which introduces the concept of "cognition-based detection" to simulate human cognition. Cefdet includes a fuzzy-driven cognitive effectiveness evaluation module (FCM), which integrates fuzzy inference into action detection and combines it with human action features to simulate the cognition-based detection process. A fuzzy cognitive update strategy (FCS) based on FCM is proposed to effectively update the results with cognitive abnormalities.

**Strengths:**

The proposed cognitive effectiveness network (Cefdet) for action detection based on fuzzy inference demonstrates several strengths as follows:
1)	Cefdet introduces the concept of "cognition-based detection" to address the limitations of existing methods in action detection. By simulating human cognition, it offers a fresh perspective on improving detection effectiveness.
2)	The paper appropriately applies fuzzy logic and fuzzy inference techniques to action detection, providing a solid technical foundation for the proposed Cefdet network. The integration of human action features further strengthens the technical correctness of the approach.
3)	The paper introduces fuzzy inference into action detection, incorporating rich input information. It takes human action features, including confidence per frame, correlation between adjacent frames, and position scores per frame, as inputs and integrates them into the fuzzy inference system.

**Limitations:**

3.	Limitations
1)	Fuzzy reasoning itself may have limitations. In the ablation experiments, it seems that the influence of threshold settings on confidence and effectiveness is limited. The portions determined through fuzzy reasoning without further processing are not validated for correctness, which could potentially impact the final action detection results. Therefore, considering the addition of validation for non-abnormal results may be necessary.
2)	In FCS, the fusion processing of low-level cognitive abilities and high-level cognitive abilities is currently limited to aggregation. However, there is room for improvement in the fusion processing approach.
3)	Simulating human cognition should be a relatively complex issue, considering only confidence makes it difficult to accurately simulate adjacent frames. Further research and exploration are needed to consider human cognition, which is also a black box problem.
4)	Adding an additional dataset to the experiment seems more convincing

**Suitability:**

2

---

### Official Review · Reviewer_gs1Z · 2024-06-01

**Rating:** 4
**Confidence:** 3

**Summary:**

The paper introduces a cognitive effectiveness network based on fuzzy inference (Cefdet) that aims to enhance the accuracy and logical consistency of action detection results by simulating human cognitive processes. To address the shortcomings of existing action detection methods that often overlook the evaluation of detection effectiveness and produce abnormal cognitive detection results, the study incorporates the concept of "cognition-based detection." The contributions of this work is the establishment of a Fuzzy-Driven Cognitive Effectiveness Evaluation Module (FCM) and a Fuzzy Cognitive Update Strategy (FCS).Combining these two modules can verify the validity of the detection results and repair abnormal cognitive detection results.

**Strengths:**

FCM comprehensively considers the confidence of each frame, the correlation between adjacent frames and the position score, uses the fuzzy inference system to accurately locate the position of the action frame with cognitive abnormalities.FCS  divides the detection results into high and low levels according to FCM, and
redetects the abnormal detection results through weight fusion of local features. FCS then uses fuzzy logic to dynamically update the results with low-level cognition before and after re-detection.
Through fuzzy logic , Cefdet achieves accurate evaluation and dynamic optimization of action detection results, solves the shortcomings of existing methods in judging and detection effectiveness, and effectively improves the accuracy and reliability of action detection.Experimental results on public datasets demonstrate Cefdet's superiority over several mainstream algorithms , which verifies its effectiveness and advantages.

**Limitations:**

While the Cefdet methodology significantly improves action detection accuracy, it is not without limitations.
1.The formulation of fuzzy inference rules is subjective, and the principle of rule setting and the process of rule optimization are not described in detail.
2.The performance of both the FCM and FCS modules is highly dependent on the quality of input features.Try to verify the effectiveness of the method in the case of some noise.
3.Although the paper highlights the effectiveness of Cefdet, it does not extensively discuss computational efficiency and resource consumption.

**Suitability:**

3

---

### Meta-Review · Area_Chair_645T · 2024-06-25

**Recommendation:** Accept (Poster)
**Confidence:** 5

**Metareview:**

All reviewers are satisfied with the responses and contributions made by the submission. The authors are suggested to take all the comments into consideration for final version.